# Population Pharmacokinetics and Probability of Target Attainment of Different Dosing Regimens of Ceftazidime in Critically Ill Patients with a Proven or Suspected *Pseudomonas aeruginosa* Infection

**DOI:** 10.3390/antibiotics10060612

**Published:** 2021-05-21

**Authors:** Annabel Werumeus Buning, Caspar J. Hodiamont, Natalia M. Lechner, Margriet Schokkin, Paul W. G. Elbers, Nicole P. Juffermans, Ron A. A. Mathôt, Menno D. de Jong, Reinier M. van Hest

**Affiliations:** 1Hospital Pharmacy and Clinical Pharmacology, Amsterdam University Medical Centre, University of Amsterdam, Meibergdreef 9, 1105 AZ Amsterdam, The Netherlands; r.mathot@amsterdamumc.nl (R.A.A.M.); r.m.vanhest@amsterdamumc.nl (R.M.v.H.); 2Medical Microbiology, Amsterdam University Medical Centre, University of Amsterdam, Meibergdreef 9, 1105 AZ Amsterdam, The Netherlands; c.j.hodiamont@amsterdamumc.nl (C.J.H.); natalia@flexxcon.com (N.M.L.); m.d.dejong@amsterdamumc.nl (M.D.d.J.); 3Clinical Pharmacy, Ziekenhuisgroep Twente, Zilvermeeuw 1, 7609 PP Almelo, The Netherlands; m.schokkin@zgt.nl; 4Amsterdam University Medical Centre, Department of Intensive Care Medicine, University of Amsterdam, Meibergdreef 9, 1105 AZ Amsterdam, The Netherlands; p.elbers@amsterdamumc.nl (P.W.G.E.); n.p.juffermans@amsterdamumc.nl (N.P.J.); 5Amsterdam University Medical Centre, Department of Intensive Care Medicine, Research VUmc Intensive Care (REVIVE), Amsterdam Medical Data Science (AMDS), Amsterdam Cardiovascular Sciences (ACS), Amsterdam Infection and Immunity Institute (AI&II), Vrije Universiteit Amsterdam, De Boelelaan 1117, 1081 HV Amsterdam, The Netherlands

**Keywords:** pharmacokinetics, pharmacodynamics, target attainment, ceftazidime, critically ill

## Abstract

Altered pharmacokinetics (PK) of hydrophilic antibiotics in critically ill patients is common, with possible consequences for efficacy and resistance. We aimed to describe ceftazidime population PK in critically ill patients with a proven or suspected *Pseudomonas aeruginosa* infection and to establish optimal dosing. Blood samples were collected for ceftazidime concentration measurement. A population PK model was constructed, and probability of target attainment (PTA) was assessed for targets 100% T > MIC and 100% T > 4 × MIC in the first 24 h. Ninety-six patients yielded 368 ceftazidime concentrations. In a one-compartment model, variability in ceftazidime clearance (CL) showed association with CVVH. For patients not receiving CVVH, variability in ceftazidime CL was 103.4% and showed positive associations with creatinine clearance and with the comorbidities hematologic malignancy, trauma or head injury, explaining 65.2% of variability. For patients treated for at least 24 h and assuming a worst-case MIC of 8 mg/L, PTA was 77% for 100% T > MIC and 14% for 100% T > 4 × MIC. Patients receiving loading doses before continuous infusion demonstrated higher PTA than patients who did not (100% T > MIC: 95% (*n* = 65) vs. 13% (*n* = 15); *p* < 0.001 and 100% T > 4 × MIC: 20% vs. 0%; *p* = 0.058). The considerable IIV in ceftazidime PK in ICU patients could largely be explained by renal function, CVVH use and several comorbidities. Critically ill patients are at risk for underexposure to ceftazidime when empirically aiming for the breakpoint MIC for *P. aeruginosa*. A loading dose is recommended.

## 1. Introduction

Ceftazidime, a third-generation cephalosporin, is a first line treatment option for critically ill patients with *Pseudomonas aeruginosa* infections. *P. aeruginosa* infections occur in critically ill patients with a reported 30-day mortality ranging between 20.9% to 49% in previous studies. These infections are typically nosocomial (ventilator associated) pneumonia, (catheter-associated) urinary tract infections or sepsis [1,2,3,4].

Early and adequate treatment of sepsis with antimicrobial therapy improves morbidity and mortality outcomes in critically ill patients with an infection [5]. Pharmacokinetics of hydrophilic antibiotics that are renally cleared, such as ceftazidime, are susceptible to variations in renal function, edema, and to the impact of resuscitation therapy during sepsis, which could lead to alterations in clearance as well as volume of distribution. In addition, the presence of co-morbidity may also influence the pharmacokinetics of these drugs [6]. These pharmacokinetic changes may cause low drug concentrations, with a risk for not achieving the pharmacokinetic/pharmacodynamic (PK/PD) target [7,8,9,10,11,12,13,14,15,16,17]. For ceftazidime, the PK/PD target in critically ill patients is reached when the free (f) drug plasma concentration is maintained above the minimum inhibitory concentration (MIC) for 100% of a dosing interval [8]. There is debate as to whether the PK/PD target should be 100% f T > MIC or whether a higher PK/PD target of 100% f T > 4 × MIC should be aimed for in critically ill patients [10].

Studies comparing different ceftazidime dosing regimens in large populations are generally lacking. Moreover, existing ceftazidime PK models are based on small study populations and these models mostly do not describe ceftazidime PK in ICU patients [9,10,11,12,13,14,15,16,17,18,19,20,21].

In this study, a population pharmacokinetic (POP/PK) analysis of ceftazidime is performed in critically ill patients with a proven or suspected *P. aeruginosa* infection. The objective is to describe the population PK of ceftazidime, quantify variability in PK between patients and to identify factors associated with this variability. Additionally, we aimed to identify the dosing regimen with optimal PK/PD target attainment. Finally, development of antimicrobial resistance was analyzed, and exploratory analyses were carried out to test whether PK/PD target attainment could be associated with microbiological and clinical cure.

## 2. Results

### 2.1. Patients and Ceftazidime Concentrations

A total of 394 blood samples were collected from 96 ICU patients. Fifteen percent of these samples were taken within the first 24 h of treatment with ceftazidime and for 46% of patients a sample was drawn within the first 24 h. The median number of samples per patient was 3 interquartile range: [1–5]. The majority of patients (83%) had a continuous intravenous dosing regimen. Only ten percent of patients were treated with an intermittent dosing regimen. The remainder switched between dosing regimens during the first 24 h of treatment. A total of 2.5% of the samples were taken during intermittent infusion. Patient characteristics are presented in Table 1. Twenty-eight (7.1%) of 394 ceftazidime samples contained a concentration below LLQ.

### 2.2. Population Pharmacokinetic Analysis

The ceftazidime data best fitted a one-compartmental model with first-order elimination (Table 2). The variability between patients, or the interindividual variability (IIV), could be estimated for CL and V. Residual variability was best described by a proportional error model. Introduction of CVVH improved the model fit as evidenced by the drop in objective function of 60.6 points (*p* < 0.001) and improvement of the goodness-of-fit plots. Further covariate analysis resulted in a model with a positive significant association between ceftazidime CL and ClCKD-EPI and associations between CL and comorbidities, indicating higher CL in the presence of the comorbidities hematologic malignancy and trauma or head injury (factor 1.57 and 1.99, respectively). The comorbidities trauma and head injury were merged into one group, due to having the same underlying mechanism for increasing drug clearance, being the hyperdynamic state with glomerular hyperfiltration. With the inclusion of these associations, the estimate for IIV CL for patients not receiving CVVH, IIV dropped from 103.4% to 36% (Table 2).

At four time points of ceftazidime sampling, CKD-EPI data were missing. Because of the small fraction of missing data, the ‘last observation carried forward’ principle was applied to handle these data. There were no missing data for the other covariates, comorbidities and CVVH. The associations between the covariates and CL are shown in Figure 1.

The final model had an adequate fit, as shown by the VPCs stratified for CVVH and non CVVH (Figure 2). Goodness-of-fit plots and the NONMEM control stream of the final model are shown in Appendix B and Appendix C, respectively.

### 2.3. PK/PD Target Attainment

For the assessment of PK/PD target attainment, 32 MIC values of isolated *P*. *aeruginosa* bacteria were available for 31 patients during 32 ICU stays. The distributions of the measured MICs are shown in Appendix A. All patients achieved the PK/PD target attainment for 100% T > MIC within the first 24 h. Of these patients, 66% (21/32) also achieved the higher target of 100% T > 4 × MIC. 

Patients receiving loading doses before continuous infusion demonstrated higher target attainment rates in the first 24 h of treatment compared to patients not receiving a loading dose for the higher target (100%T > 4 × MIC: 72% (*n* = 25) vs. 0% (*n* = 4); *p* = 0.006).

PK/PD target attainment was also calculated for all patients treated with ceftazidime ≥1 day, using the worst case (breakpoint)MIC of 8 mg/L, which is a realistic scenario, as ceftazidime is used as empirical therapy in the treatment of suspected *P*. *aeruginosa* infections when no MIC is available yet. This could be estimated for 94 patients with 96 treatment periods longer than 24 h. PK/PD target was achieved in 77% of the patients for the target of 100% T > MIC, and 14% achieved the target of 100% T > 4 × MIC. Administration of a loading dose before continuous infusion resulted in higher PK/PD target attainment for both PK/PD targets within the first 24 h of treatment [100%T > MIC: 95% (*n* = 65) vs. 13% (*n* = 15); *p* < 0.001 and 100%T > 4 × MIC: 20% vs. 0%; *p* = 0.058].

### 2.4. Monte Carlo Dosing Simulations

The association between ceftazidime clearance and CLCKD-EPI is illustrated in the simulated concentration–time profiles for the dosing regimen 3 g continuous infusion and 2 g loading dose with 5 g continuous infusion. For this, the median CLCKD-EPI and both the 10th and 90th percentile of the study population was used. Figure 3a,b show that patients with higher CLCKD-EPI had lower ceftazidime concentrations. For *P*. *aeruginosa* infections with an MIC of 8 mg/L and patients with a CLCKD-EPI of 122 mL/min, simulations with a 3 g continuous infusion and a 2 g loading dose followed byy 5 g continuous infusion regimen showed that 10.8% and 97.9%, respectively, achieved the 100%T > MIC target. In Figure 3c,d, the concentration–time profile with the same two dosing regimens is shown for the different comorbidities. For *P*. *aeruginosa* infections with an MIC of 8 mg/L, and patients with the comorbidity ‘trauma or head injury’, 100%T > MIC was achieved in 9.1% and 97.9%, respectively, for the 3 g continuous infusion and the 2 g loading dose followed by 5 g continuous infusion.

Furthermore, PTA was calculated for frequently applied dosing regimens and different MICs (Figure 4). For *P*. *aeruginosa* infections with an MIC of 8 mg/L, simulations showed that the PTA of 2 g loading dose and 5 g continuous infusion regimen was 98.4% for 100%T > MIC and 65.6% for 100%T > 4 × MIC. For a continuous dosing regimen without a loading dose, PTA did not exceed 40% of the simulated patients with a *P*. *aeruginosa* infection with an MIC of 8 mg/L for both 100%T > MIC and 100%T > 4 × MIC.

### 2.5. Clinical Outcome Measures: Microbiological and Clinical Cure

For 17 patients, the endpoint microbiological cure could be assessed. Of these, 9 (53%) patients had isolates which became resistant (category C in Table 3) during therapy. In this study, only one negative follow-up isolate (category A in Table 3) was identified.

For 21 patients, the endpoint clinical cure could be assessed. Ten (48%) patients achieved clinical cure during treatment with ceftazidime. Eleven patients failed on treatment with ceftazidime, meaning they were escalated to other anti-*P*. *aeruginosa* therapy. Among the patients with clinical failure and of whom a microbiological outcome was known (*n* = 10), 80% developed decreased susceptibility (category C).

No association could be found between PK/PD target attainment and the clinical outcome measures.

## 3. Discussion

In the present study, a population PK model of ceftazidime in adult ICU patients with a suspected or proven *P. aeruginosa* infection was developed. The study population was generally severely ill, as illustrated by the median SOFA score of 10. A one-compartment model best described the ceftazidime PK. The CLnonCVVH was comparable to the values found in previous studies [16,18]. However, V was nearly two-fold higher than found in previous studies [10,12]. A possible explanation could be that patients in the current study where more severely ill, as indicated by the SOFA score. Additionally, in the previous studies, V was estimated for patients receiving intermittent dosing, whereas in our study, continuous dosing was mostly used. 

The interpatient variability of ceftazidime PK was high, for example, it was 103.4% in CLnonCVVH in the structural model. This variability could largely be explained when creatinine clearance (CLCKD-EPI) was taken into account. Since ceftazidime is a hydrophilic drug with low protein binding and with predominant renal clearance, this is an expected finding. Furthermore, the comorbidities hematologic malignancy, trauma or head injury explained variability on CLnonCVVH. These comorbidities have been shown to cause augmented clearance of other hydrophilic antibiotics in previous studies [22,23,24]. 

Although there was a large drop in IIV CL in the final model relative to the structural model upon inclusion of the covariates (from 103.6% to 36.0%), there was a simultaneous increase in IIV V (from 84.7% to 102.8%). An explanation might be that the vast majority of patients received continuous infusion, making it more difficult to separate the IIV that belongs to CL from the IIV that belongs to V than in situations where greater data availability from intermittent infusion. Importantly, overall variability decreased with the addition of the covariates.

This study showed that critically ill patients with *P. aeruginosa* infections are at considerable risk for underexposure to empirical therapy with ceftazidime in the first 24 h of treatment, when a worst-case MIC for *P. aeruginosa* of 8 mg/L needs to be covered (77% and 14% achieved the targets of 100% T > MIC and 100% T > 4 × MIC, respectively). This is reason for concern. The risk of not attaining the target was especially high when a loading dose was omitted. In addition, there is a high risk of not attaining the target when the higher target of 100%T > 4 × MIC was aimed for (66% of included patients in whom a baseline MIC was available (*n* = 32) achieved this target). On the other hand, these patients all achieved the target 100% T > MIC.

Monte Carlo simulations gave further insight into the influence of different dosing regimens and the identified covariates on PTA. The probability of PK/PD target attainment was lower with higher CLCKD-EPI and in the presence of the defined comorbidities when a 3 g continuous infusion dosing regimen was applied and when 100%T > 4 × MIC was aimed for with worst-case MIC (Figure 2a,c). When 5 g per 24 h continuous infusion with a 2 g loading dose was simulated, the PTA was barely affected by changes in CLCKD-EPI or the presence of comorbidities (Figure 2b,d). Furthermore, simulations of different dosing regimens showed that less than 50% of patients treated with a continuous dosing regimen without loading doses achieve the PK/PD targets when treating *P. aeruginosa* infections with MICs of 4 mg/L and above. Since our PK model is based for the most part on concentration–time data from patients receiving continuous infusion, the model was used to simulate continuous dosing regimens only.

In this study, the relationship between ceftazidime concentrations and toxicity was not investigated. In general, ceftazidime is a drug with relatively low toxicity. However, neurotoxicity has been reported in patients with renal failure, elderly, and patients with neurological disorders [25]. Although a concentration cut-off for toxicity is not known, therapeutic drug monitoring could be used in patients with high risk for developing this adverse event. The dose should be adjusted when the ceftazidime concentration is far above the target needed for effect against *P. aeruginosa.* The adaptive target of C < 10 × MIC (<80 mg/L in empiric therapy) could be used as proposed by Gatti et al. [25].

To our knowledge, this is the first study to investigate the association between ceftazidime target attainment and microbiological and clinical cure. In 53% of the patients in which follow-up isolates were available, *P. aeruginosa* developed resistance for ceftazidime during therapy. Only one of these patients was classified as achieving microbiological cure. This observation is prone to selection bias, as follow-up isolates are more likely to be collected in patients who are not recovering. However, even in comparison with the total population studied (*n* = 96, i.e., best case scenario), the incidence of development of resistance is high (almost 10%), confirming the findings of earlier studies [26,27]. No statistically significant difference was observed in ceftazidime target attainment between patients with and without development of microbiological resistance, yet the numbers per group were small (*n* = 9 and 8, respectively). There was also no observed statistically significant association between ceftazidime target attainment and clinical cure, and again likely due to small patient numbers (*n* = 10 and 11, respectively).

This study has several limitations. Firstly, our results could be influenced by selection bias. Cultures were only taken on clinical indication and the follow-up of patients varied as a result of the observational design of our study. Consequently, the patients with more cultures available could be more severely ill. Therefore, the percentage of patients with microbiological failure was probably overestimated since patients with no follow-up isolates were excluded from that part of the analysis. 

Secondly, although our study included a high number of patients and ceftazidime samples for the primary aim of the study, being the assessment of the population PK of ceftazidime in ICU patients, the sample size for the secondary aims, being exploration of associations with clinical outcomes, was limited.

Thirdly, we used the CKD-EPI formula for the estimation of renal clearance, which has limited predictive value in critically ill patients [28]. However, use of the CKD-EPI resulted in a better fit of the model compared to the use of the AKIN score.

Fourthly, since molecular analysis of resistant *P. aeruginosa* strains was not performed, there was no further insight into the underlying mechanisms of the resistance pattern.

Fifthly, only 15% of the collected blood samples were obtained within the first 24 h of treatment. Therefore, one could argue that the developed model might not be suitable to calculate the target attainment within the first 24 h. However, during the development of the model, interoccasion variability for both V and CL was tested and found not to improve the fit of the model. Therefore, no significant difference in PK between different days, other than that accounted for by CLCKD-EPI and CVVH, could be identified.

Finally, this study was carried out in a single center. Therefore, the results that were found might not be representative for other hospitals.

## 4. Materials and Methods

### 4.1. Study Design and Setting

The current study was an observational population pharmacokinetic study of ceftazidime at the ICU of Amsterdam University Medical Center, location AMC, a tertiary referral center in Amsterdam, The Netherlands. The institutional review board of the Amsterdam University Medical Center considered the study as not requiring WMO approval. Patients and relatives were given the opportunity for an opt-out consent method.

### 4.2. Study Population

ICU Patients aged ≥18 years treated with IV ceftazidime for a proven or suspected clinically relevant *P. aeruginosa* infection, and with at least one detectable ceftazidime serum concentration available during the course of therapy, were included. Cystic fibrosis patients were excluded. If patients received ceftazidime therapy after discontinuation for more than 28 days, this was assessed as a new treatment period. 

For the secondary objective, PTA, the inclusion criterion was one positive *Pseudomonas aeruginosa* culture with a successful MIC measurement for calculation of PK/PD target attainment and treatment for at least 24 h.

Microbiological and clinical cure were evaluated for patients in whom PK/PD target attainment could be calculated. For the assessment of microbiological cure, the availability of at least one follow-up culture from a relevant location was needed with successful MIC measurement >48 h while the patient was receiving ceftazidime treatment. Additionally, the treatment period with ceftazidime had to be longer than 48 h for inclusion. For both microbiological and clinical cure, patients receiving anti-pseudomonal agents for treatment of a different suspected or proven infection than for which the ceftazidime course was prescribed were excluded.

### 4.3. Outcome Measures

In this study, several outcome measures were evaluated. The primary objective of this study was to develop a ceftazidime population pharmacokinetic (POP/PK) model in critically ill patients using nonlinear mixed effect modelling (NONMEM) and to quantify and explain the interpatient variability in ceftazidime exposure. As such, primary outcome measures are the population PK parameters and the variability in these parameters.

Secondary outcome measures were (i) PK/PD target attainment, (ii) microbiological cure, and (iii) clinical cure. The definitions of these endpoints are displayed in Table 3.

### 4.4. Sample and Data Collection

Ceftazidime samples were obtained prospectively, as part of routine clinical care, from both waste materials of arterial blood gas samples, assuring random sampling and from routine therapeutic drug monitoring (TDM), for which samples were collected at standard rounds in the early morning on every Monday, Wednesday and Friday. PK data were collected from ICU patients admitted between November 2013 and March 2018. 

Baseline patient characteristics and ceftazidime treatment data were retrieved retrospectively from the Patient Data Monitoring System (PDMS) Metavison (iMDsoft, Tel Aviv, Israel) and EPIC (EPIC Systems Corporation, Verona, WI, USA). Over the years of the study, different ceftazidime dosing regimens have been applied on the ICU for the treatment of proven or suspected infections with *P. aeruginosa*. These dosing regimens ranged from intermittent dosing of 1 g tid or 2 g tid, to 3 g or 6 g over 24 h via continuous infusion, with and without loading doses. A loading dose was defined as a bolus administered in several minutes immediately before initiation of continuous infusion.

The following data were collected: admission type, time, dose and administration mode (intermittent or continuous) of ceftazidime administration, time of sample collection, sex, age, bodyweight, BMI, height, Sepsis-related Organ Failure Assessment (SOFA) score at the start of ceftazidime treatment, Acute Kidney Injury Network (AKIN) score, and comorbidities including hematologic malignancy, oncologic malignancy, acute trauma, and head injury. During treatment, the serum creatinine, estimated glomerular filtration rate (eGFR, calculated with the Chronic Kidney Disease Epidemiology Collaboration (CKD-EPI) Equation 2009), serum albumin, serum sodium and use of continuous veno-venous hemofiltration (CVVH) and mechanical ventilation, were obtained. 

Furthermore, information on norepinephrine use and furosemide use during ceftazidime therapy was collected. Missing data in these patients were replaced with the closest value in time, or when absent, the median population value.

Measured MIC values from the positive *Pseudomonas aeruginosa* cultures were used for the assessment of attaining the PK/PD target. In addition, PK/PD target attainment was calculated by using a surrogate worst case MIC of 8 mg/L for *Pseudomonas aeruginosa*, being both the highest MIC within the wildtype distribution and the breakpoint, since measured MIC data were not available for all patients. This MIC was extracted from the European Committee on Antimicrobial Susceptibility Testing (EUCAST) database [29].

### 4.5. Drug Assay and Isolates

Serum blood samples were centrifuged and stored at −80 °C, in the pharmacy’s research laboratory. Since protein binding for ceftazidime is low (approximately 10%), total serum concentrations were measured. These concentrations were measured using a validated high-performance liquid chromatography tandem mass spectrometry (LC-MS/MS) method (LC:LC30 Shimadzu, Kyoto, Japan; MS QTRAP 5500 system, Sciex, Framington, MA, USA). The lower limit of quantification (LLQ) was 0.5 mg/L and the higher limit of quantification (HLQ) was 40 mg/L. Concentrations higher than the HLQ were reanalyzed after dilution. Accuracy at concentrations of 0.5, 10 and 40 mg/L was 106.2%, 102.2% and 102.2%, respectively. Precision at concentrations of 0.5, 10 and 40 mg/L was 109.8%, 92.4% and 102.4%, respectively. 

In this study, all *Pseudomonas aeruginosa* isolates from samples taken for clinical purpose were collected. Identification was performed by MALDI-TOF MS (Bruker Daltonics, Billerica, MA, USA). The MICs of ceftazidime for the *P. aeruginosa* isolates were determined semi-automatically using the VITEK 2 system (BioMerieux, Marcy-l’Étoile, France) or manually by E-test (BioMerieux), carried out by the department of Medical Microbiology at the AMC.

### 4.6. Population Pharmacokinetic Analysis

POP/PK analysis was performed using nonlinear mixed-effects modeling software (NONMEM 7.1.2; Icon Development Solutions, Ellicott City, MD, USA). Detailed methodologic information on model development and validation is available in Appendix D.

### 4.7. Monte Carlo Simulations

The final POP/PK model was used to simulate ceftazidime concentration–time curves for the dosing regimens 3 g continuous infusion and 2 g loading dose with 5 g continuous infusion, to generate insight in the magnitude of the effect of the identified covariates on ceftazidime exposure. The concentration–time curves following these dosing regimens were simulated for the first 24 h of treatment for 1000 virtual patients with all median characteristics of the population but with the 10th, 50th and 90th percentile values of the statistically significantly associated covariates from the final model.

To generate insight into PTA, the percentage of patients expected to attain 100% T > MIC and 100% T > 4 × MIC in the first 24 h of treatment was calculated for different dosing regimens: 3 g via continuous infusion with or without a loading dose and 5 g via continuous infusion with or without a loading dose, which are the most frequently applied dosing regimens at our ICU. Simulations of these dosing regimens were performed for patients with all median characteristics of the population. One thousand virtual patients were simulated for each dosing regimen, and target attainment was calculated for different MICs, ranging from 1 to 8.

### 4.8. Statistical Analysis

Data are presented as percentages for categorical values and median values and ranges for continuous variables. Differences in PTA between patients with different dosing regimens were compared using the Pearson chi square test. A two-sided *p*-value of <0.05 was considered statistically significant. Statistical analysis was performed using IBM SPSS Statistics v25 (IBM corporation, Armonk, NY, USA).

## 5. Conclusions

In conclusion, the population PK of ceftazidime in critically ill patients with a suspected or proven *P. aeruginosa* infection demonstrated a high interindividual variability, which could to a large extent be explained by CLCKD-EPI, CVVH and the comorbidities hematologic malignancy and trauma or head injury. 

Critically ill patients are at risk of underexposure to ceftazidime, in particular, in the case of infections with an increased MIC. A loading dose prior to continuous infusion dosing regimens improved PTA. These results are in line with the performed simulations, suggesting that a dosing regimen of a 2 g loading dose followed by 5 g via continuous infusion can be recommended for optimal target attainment. Development of resistance of *P. aeruginosa* against ceftazidime seems common during therapy with ceftazidime.

## Figures and Tables

**Figure 1 antibiotics-10-00612-f001:**
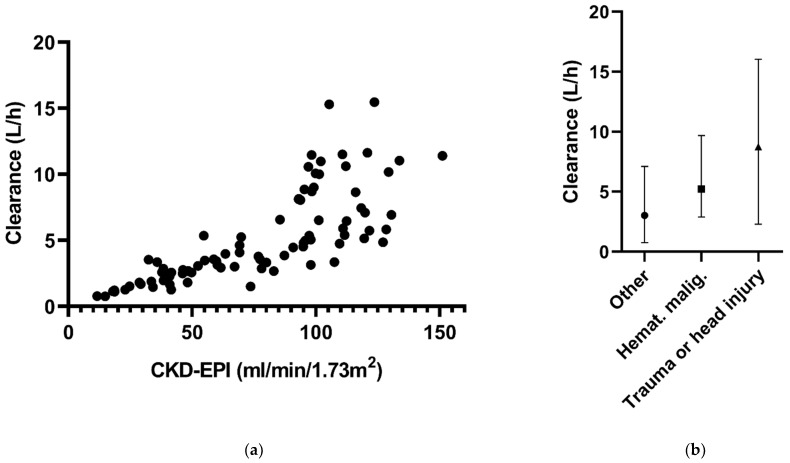
Ceftazidime clearance in relation to (**a**) CKD-EPI and (**b**) the different comorbidities in the final model.

**Figure 2 antibiotics-10-00612-f002:**
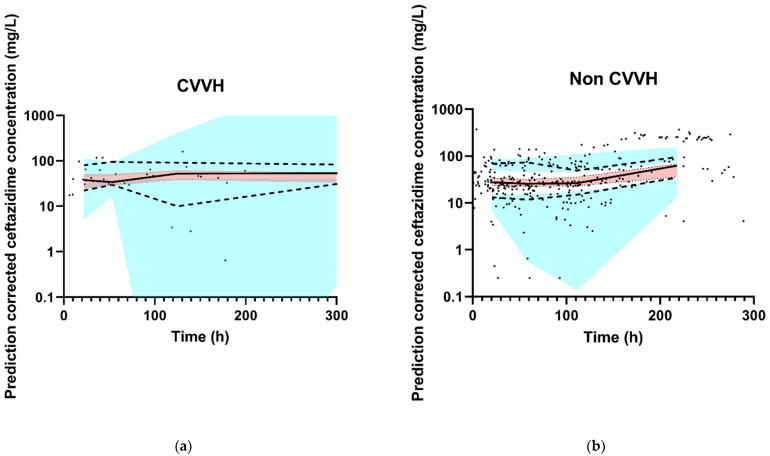
Observed ceftazidime concentration–time data and prediction-corrected VPC of the final model. The black dots represent the observed ceftazidime concentrations. The thick red line is the observed median, and the small blue lines are the 5th and 95th percentiles of the observed data. The red shaded area represents 95% CI of the model-predicted median and the blue-shaded areas are the 95% CIs of the model-predicted 5th and 95th percentiles. (**a**) CVVH patients, and (**b**) non CVVH patients. For the *X*-axis, the VPC was zoomed in on the first 300 h in order to properly assess the fit. For both groups (non CVVH and CVVH), 12 data points were collected after 300 h and are therefore not in the figures. The thick red line and small blue lines run within their shaded areas, demonstrating an adequate fit of the model.

**Figure 3 antibiotics-10-00612-f003:**
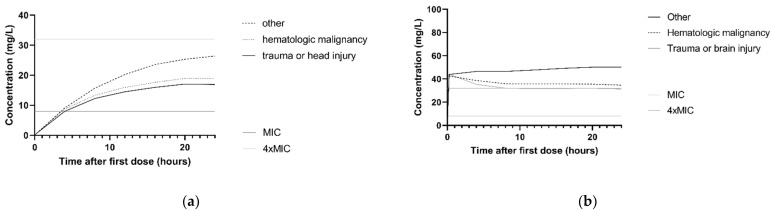
Simulations of ceftazidime concentration–time profiles in the first 24 h. The median of *n* = 1000 virtual patients is shown. The MIC and 4 × MIC lines are displayed for a worst-case MIC of 8 mg/L. (**a**) Simulation of 3 g continuous infusion dosing regimen for patients with a CLCKD-EPI of 33 mL/min/m^2^ (10th percentile), 73 mL/min/m^2^ (median) and 122 mL/min/m^2^ (90th percentile). All patients were simulated with the comorbidity ‘other’. (**b**) Simulation of 2 g loading dose followed by 5 g continuous infusion for patients with the a CLCKD-EPI of 33 mL/min/m^2^ (10th percentile), 73 mL/min/m^2^ (median) and 122 mL/min/m^2^ (90th percentile). All patients were simulated with the comorbidity ‘other’ (**c**) Simulation of 3 g continuous infusion dosing regimen for patients with different comorbidities: other, hematologic malignancy and trauma or head injury. All patients were simulated with a median CLCKD-EPI. (**d**) Simulation of 2 g loading dose with 5 g continuous infusion for patients with different comorbidities: other, hematologic malignancy and trauma or head injury. All patients were simulated with a median CLCKD-EPI.

**Figure 4 antibiotics-10-00612-f004:**
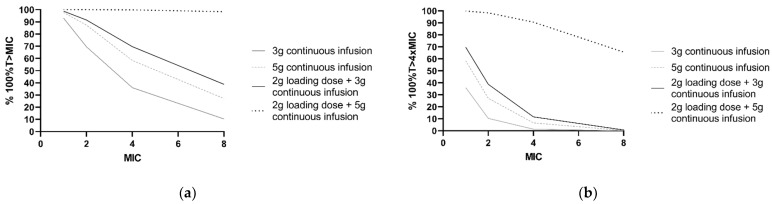
PK/PD target attainment in the first 24 h with four different ceftazidime dosing regimens. Percentages of 1000 patients simulated with a median CL_CKD-EPI_ of 73 mL/min/m^2^ and ‘other’ comorbidity. (**a**) For achievement of 100%T > MIC and (**b**) 100%T > 4 × MIC for a range of MIC values. The clinical susceptibility breakpoint for *P. aeruginosa*, according to the European Committee on Antimicrobial Susceptibility Testing, is 8 mg/L.

**Table 1 antibiotics-10-00612-t001:** Baseline characteristics of critically ill patients at ceftazidime therapy initiation (*n* = 96).

Characteristic	Median [Range]
Female, *n* (%)	38 (40%)
Age, yrs	59 [20–84]
Body weight, kg	79 [44–237]
Body mass index, kg/m^2^	25 [16–66]
Ceftazidime dose prescribed in the first 24 h, *n* (%)	
1 g tid	7 (7%)
2 g tis	3 (3%)
<3 g continuous infusion	1 (1%)
3 g continuous infusion	34 (35%)
3–5 g continuous infusion	11 (11%)
5 g continuous infusion	25 (25%)
6 g continuous infusion	9 (9%)
Other	6 (%)
Loading dose, *n* (% of patients with continuous infusion)	65 (81%)
SOFA score at start of ceftazidime therapy (*n* = 64) ^c^	10 [4–16]
30-day mortality, *n* (%)	37 (39%)
Primary infection site, *n*(%)	
Pneumonia	37 (39%)
Bloodstream	17 (18%)
Abdominal infection	13 (14%)
Meningitis	22 (23%)
Other	7 (7%)
Admission Category, *n* (%)	
Medical	54 (56%)
Surgical	42 (44%)
Comorbidity, *n* (%)	
Hematologic malignancy	14 (15%)
Oncologic malignancy	12 (13%)
Trauma or head injury	27 (28%)
Other	43 (45%)
Vasopression, *n* (%)	66 (69%)
Ventilation, *n* (%)	74 (77%)
Creatinine, mg/dL	0.98 [0.19–7.49]
eGFR ^a^, mL/min/m^2^	73 [6–153]
CVVH, *n* (%) ^b^	20 (21%)
RIFLE score, *n* (%)	
No AKI	62
Stage 1	4
Stage 2	1
Stage 3	29
Mean Inhibitory concentration (mg/L) *P. aeruginosa* at start therapy, *n* (%)	
1	6 (19%)
2	13 (40%)
4	8 (25%)
8	3 (9%)
16	2 (6%)

^a^ The estimated glomerular filtration rate is calculated using the Chronic Kidney Disease Epidemiology Collaboration (CKD-EPI) formula. ^b^ Patients with application of CVVH during (a part of) their treatment with ceftazidime. ^c^ SOFA, Sequential Organ Failure Assessment. The SOFA score could be assessed for only 64 patients because of missing data in, for example, the Glasgow coma scale.

**Table 2 antibiotics-10-00612-t002:** Parameter estimates of the structural and final model.

	Structural Model		Final Model		Bootstrap ^#^	
	Estimation	RSE (%)	Estimation	RSE (%)	Estimation	95% CI
CL CVVH (L/h)	2.82	11	2.9	11	2.88	2.18–3.47
CL non CVVH (L/h)	4.56	9	3.42	9	3.46	2.88–4.04
V (L)	47.6	13	46.8	12	46.7	37.5–59.5
Proportional error	0.288	12	0.281	12	0.277	0.216–0.352
IIV						
CL non CVVH (CV%)	103.4	11	36.0	14	35.3	24.7–46.8
V (CV%)	84.7	15	102.8	18	100.1	59.8–160.0
Covariate effects						
CKD-EPI	-	-	0.772 ^a^	11	0.788	0.655–1.022
Comorbidity hematologic malignancy	-	-	1.57	17	1.54	1.07–2.15
Comorbidity trauma or head injury	-	-	1.99	13	1.96	1.51–2.55

Abbreviations: CI, confidence interval; CL, clearance; CV%, variation coefficient in %; IIV, interindividual variability; RSE, relative standard error; V, volume of distribution. The shrinkage was 29% for both IIV on CL and V. ^a^ CLnonCVVH = 3.42 * (CKD-EPI individual/median CKD-EPI population)^0.772^ * 1.57(hemat) * 1.99(trauma/head injury), hemat = 1 if comorbidity was hematologic malignancy, zero if otherwise. Trauma/head injury = 1 if comorbidity was trauma or head injury, zero if otherwise. ^#^ 98.2% of bootstrap runs were successful. The condition number for the final model was 19.81, indicating that the model was stable.

**Table 3 antibiotics-10-00612-t003:** Definitions of the secondary endpoints.

Secondary Endpoints	Definition
100% T > MIC	Ceftazidime concentration maintained above MIC of the pathogen throughout ≥95% of the first 24 h of treatment.
100% T > 4 × MIC	Ceftazidime concentration maintained above a concentration 4-fold higher than the MIC of the pathogen throughout ≥95% of the first 24 h of treatment.
Microbiological response: assessed between 48 h after start of therapy until 48 h after stop of therapy	
Patients with microbiological cure.	*P. aeruginosa* cultures become negative during or after ceftazidime treatment.
Patients with microbiological failure without decreased susceptibility for ceftazidime.	*P. aeruginosa* cultures (from the same or relevant location) remain positive during ceftazidime treatment, MIC remains equal.
Patients with microbiological failure with decreased susceptibility (resistance) for ceftazidime.	*P. aeruginosa* cultures (from the same or relevant location) remain positive during ceftazidime treatment, MIC increases with at least factor 4.
Clinical response	
Clinical cure	Completion of full treatment course without change or addition of antibiotic therapy, and no additional antibiotics commenced within 48 h of cessation.
Clinical failure	Any clinical outcome other than clinical cure.

## Data Availability

Data available on request due to restrictions: privacy.

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
