# Peer review of "Population Pharmacokinetics and Probability of Target Attainment of Different Dosing Regimens of Ceftazidime in Critically Ill Patients with a Proven or Suspected *Pseudomonas aeruginosa* Infection"

_antibiotics, 2021, doi:10.3390/antibiotics10060612_

Round 1

Reviewer 1 Report

The authors conducted PPK analysis of ceftazidime in ICU patients infected by P. aeruginosa, and recommended the regimen including loading dose to achieve the PK/PD target for the efficacy. This study is important to promote an optimal dosing regimen against prevention of resistant pathogens. Although the study is worth publishing in this journal, the authors should revise some parts.

Major

  1. Based on clinical dosing regimen (2g×3/day), CLSI shows MIC of ceftazidime against aeruginosa. Moreover, a general dosing regimen is similar with the clinical dosing regimen. However, the authors recommended total 7g/day including 2g loading dose. The recommended regimen is non-clinical.
  2. Then, why were the continuous infusion regimens high percentage in this study? Is the continuous infusion common in the world?
  3. When the authors recommend the optimal dosing regimen, results to analysis adverse events are needed. Therefore, the authors should add the results regarding dosing regimens to reduce adverse events.
  4. The equations of final model are important in this study. Please describe the equations.
  5. Table 2 and Appendix A: The CV%s of V in the structural and final model show over 50%. Moreover, an x-axis in the figure of population predicted concertation versus CWRES shows a few points in the range of less 0. Why are there a few points in the range?
  6. The authors should add a figure regarding a profile of serum ceftazidime concentration versus time and MIC distribution for aeruginosa for ceftazidime, because readers are easy to understand the population.

Minor

  1. Table 1: What is “Other” in the section of ceftazidime dose prescribed in the first 24 hours? The authors should describe all regimens.
  2. Table 1: Please change from micromole/L to mg/dL regarding a unit of Creatinine.
  3. Table 2: What is IIV? Please describe IIV in method.
  4. Figure 2: Are the title correct? I thought that (a) and (b) show “CVVH” and “No CVVH”, respectively.

Author Response

The authors conducted PPK analysis of ceftazidime in ICU patients infected by P. aeruginosa, and recommended the regimen including loading dose to achieve the PK/PD target for the efficacy. This study is important to promote an optimal dosing regimen against prevention of resistant pathogens. Although the study is worth publishing in this journal, the authors should revise some parts.

Thank you for reviewing this manuscript. The comments are provided with our answers below. The adjustments we made in the manuscript are available in the attachment.

Major

Point 1. Based on clinical dosing regimen (2g×3/day), CLSI shows MIC of ceftazidime against aeruginosa. Moreover, a general dosing regimen is similar with the clinical dosing regimen. However, the authors recommended total 7g/day including 2g loading dose. The recommended regimen is non-clinical.

For ceftazidime, the PK/PD target in critically ill patients is reached when the free (f) drug plasma concentration is maintained above the MIC for 100% of a dosing interval. Theoretically a better %T>MIC is achieved with continuous infusion. Therefore, betalactams are dosed more and more via continuous infusion. This is also recommended in our national SWAB policy:

‘In patients with sepsis we suggest prolonged or continuous* infusion

of other beta-lactam antibiotics than piperacillin-tazobactam and

carbapenems’ [1]

  1. Dutch Working Party on Antibiotic Policy. SWAB guideline for empirical antibacterial therapy of sepsis in adults. 2020. Available from: www.swab.nl/en/sepsis cited on 6 may 2021

Point 2. Then, why were the continuous infusion regimens high percentage in this study? Is the continuous infusion common in the world?

The percentage continuous infusion is high because the protocol in the hospital changed from intermittent to continuous dosing.

Point 3. When the authors recommend the optimal dosing regimen, results to analysis adverse events are needed. Therefore, the authors should add the results regarding dosing regimens to reduce adverse events

Ceftazidime is a drug with relatively low toxicity. However, some studies report neurologic toxicity in patients with renal failure. A relationship between concentration and this adverse event is not (yet) established. TDM could be used and the dose could be adjusted when the concentration is far above the target for effectivity. Gatti et al. proposed C<10xMIC.This consideration is added in the discussion. (line 240-247, page 9)

This study was not designed for the assessment of relationship between concentrations or covariates and toxicity. Furthermore, research after these adverse events would probably need a larger study population since these events occur rarely.

Point 4. The equations of final model are important in this study. Please describe the equations.

The equation is added in the subscript of table 2:

CLnonCVVH=3.42 * (CKD-EPI individual / median CKD-EPI population)0.772 * 1.57(hemat) * 1.99(trauma/head injury), hemat=1 if comorbidity was haematologic malignancy, zero if otherwise. Trauma/head injury=1 if comorbidity was trauma or head injury, zero if otherwise.

(line 105, page 5)

Point 5.

a Table 2 and Appendix A: The CV%s of V in the structural and final model show over 50%.

  1. b. Moreover, an x-axis in the figure of population predicted concertation versus CWRES shows a few points in the range of less 0. Why are there a few points in the range?

For a hydrophilic antibiotic variability in V is expected in critically ill patients. However, a significant association with the tested covariates was not found. With the large decrease in IIV for CLnoncvvh comparing the structural model to the final model, an increase in IIV for V occurred. An explanation might be that the vast majority of patient received continuous infusion, making it more difficult to separate the IIV that belongs to CL from the IIV that belongs to V than in a situation where more data would have been available from intermittent infusion. Importantly, the overall variability decreased. (line 213)

The x-axis in the figure of population predicted concentration vs CWRES shows a few points below 0 because the data in our dataset were logarithmically transformed. The points below 0 are concentrations between 0 and 1 mg/L. (line 520, page 18)

Point 6. The authors should add a figure regarding a profile of serum ceftazidime concentration versus time and MIC distribution for aeruginosa for ceftazidime, because readers are easy to understand the population.

The ceftazidime concentrations versus time are shown in figure 2. This is a VPC and also shows the concentrations. (line 121 page 6)

A figure showing the MIC distribution was added in appendix A. (line 512, page 17)

Minor

Point 1. Table 1: What is “Other” in the section of ceftazidime dose prescribed in the first 24 hours? The authors should describe all regimens.

The ‘other’ dosing regimen group contained patients who were switched between 2 regimen within the first 24h and therefore were administered a different dosage compared to the standard dosing groups.

Point 2. Table 1: Please change from micromole/L to mg/dL regarding a unit of Creatinine.

This is adjusted as proposed by the reviewer. (line 81, page 3)

Point 3. Table 2: What is IIV? Please describe IIV in method.

IIV stands for interindividual variability. Elucidation is added in the material and methods (supplemental material) and in also the results section. (line 88, page 4 and line 579, page 21)

Point 4. Figure 2: Are the title correct? I thought that (a) and (b) show “CVVH” and “No CVVH”, respectively.

‘a. noCVVH and b CVVH’ are now changed in bold fond style for clarification. (line 124, page 6)

Reviewer 2 Report

In this study, the author performed PPK analysis of ceftazidime in critically ill patients. This paper seemed useful for treatment of P. aeruginosa infection. However, there are several points should be revised for publication.

Table I The creatinine clearance is calculated using the CKD-EPI formula. However, The CKD-EPI formula is used to calculate GFR. Therefore, it would be incorrect to use the CKD-EPI formula to calculate CCr.

Fig.1 The author showed the relationship between CKD-EPI and Clearance. However, the author did not explain how to determine the clearance of Ceftazidime in each patients. The author must explain the method for calculation the clearance.

Fig.3 The author divided the patients by eGFR. Please explain why the author divided 122, 73 and 33 mL/min/m2.

Fig.3 Is the continuous infusion of Ceftazidime general in clinical? In the table I, there are several methods for administration of Ceftazidime. Why the author did not simulate the 1g tid and 2 g tis.

Author Response

In this study, the author performed PPK analysis of ceftazidime in critically ill patients. This paper seemed useful for treatment of P. aeruginosa infection. However, there are several points should be revised for publication.

Thank you for the encouraging comment. The points below are processed in order to improve the paper.

Point 1. Table I The creatinine clearance is calculated using the CKD-EPI formula. However, The CKD-EPI formula is used to calculate GFR. Therefore, it would be incorrect to use the CKD-EPI formula to calculate CCr.

CCr is changed in GFR as proposed by the reviewer. (line 81, page 3)

Point 2. Fig.1 The author showed the relationship between CKD-EPI and Clearance. However, the author did not explain how to determine the clearance of Ceftazidime in each patients. The author must explain the method for calculation the clearance.

The formula for calculating the clearance is added in the subscript of table 2:

CLnonCVVH=3.42 * (CKD-EPI individual / median CKD-EPI population)0.772 * 1.57(hemat) * 1.99(trauma/head injury), hemat=1 if comorbidity was haematologic malignancy, zero if otherwise. Trauma/head injury=1 if comorbidity was trauma or head injury, zero if otherwise. (line 105, page 5)

Point 3. Fig.3 The author divided the patients by eGFR. Please explain why the author divided 122, 73 and 33 mL/min/m2.

These values are the median, 10th percentile and 90th percentile for eGFR in the study population. These were chosen to show the variability  This elaboration is added in the manuscript. (line 150, page 7)

Point 4. Fig.3 Is the continuous infusion of Ceftazidime general in clinical? In the table I, there are several methods for administration of Ceftazidime. Why the author did not simulate the 1g tid and 2 g tis.

The proposed simulation of 1g tid and 2g tid was part of the original plan. However, the model in this study was based on a study population with only 10% intermittent dosing. Therefore we felt it would not be correct to simulate intermittent dosing regimen using this model.

For ceftazidime, the PK/PD target in critically ill patients is reached when the free (f) drug plasma concentration is maintained above the MIC for 100% of a dosing interval. Theoretically a better %T>MIC is achieved with continuous infusion. Therefore, betalactams are dosed more and more via continuous infusion. This is also recommended in our national SWAB policy:

‘In patients with sepsis we suggest prolonged or continuous* infusion

of other beta-lactam antibiotics than piperacillin-tazobactam and

carbapenems’ [1]

  1. Dutch Working Party on Antibiotic Policy. SWAB guideline for empirical antibacterial therapy of sepsis in adults. 2020. Available from: www.swab.nl/en/sepsis cited on 6 may 2021

Round 2

Reviewer 1 Report

The authors revised appropriately. No further correction is necessary.

Author Response

No further correction for reviewer 1 was performed.

Reviewer 2 Report

The author revised the manuscript according to reviewer's comments. However there are several points should be revised.

#1 P3 line80: The CCr is not changed in GFR.

#2 Fig1: The author showed the relationship between CKD-EPI and Clearance. Ceftazidime is mainly excreted by the kidneys, so it is not surprising to see such a correlation. Where is the novelty in this data?

Author Response

The author revised the manuscript according to reviewer's comments. However there are several points should be revised.

Point 1. P3 line80: The CCr is not changed in GFR.

This is now corrected. (line 80, page 3)

Point 2. Fig1: The author showed the relationship between CKD-EPI and Clearance. Ceftazidime is mainly excreted by the kidneys, so it is not surprising to see such a correlation. Where is the novelty in this data?

It was indeed already known that ceftazidime is mainly excreted by the kidneys. However, the description of the impact of CKD-EPI on the population PK of ceftazidime, and on the probability of target attainment in critically ill patients is new. In this study, we showed the clinical effect by quantifying the effect of renal function,